# The activity of human enhancers is modulated by the splicing of their associated lncRNAs

**Jennifer Yihong Tan** *, **Ana Claudia Marques** *

Department of Computational Biology, University of Lausanne, Lausanne, Switzerland

* jennifer.tan@unil.ch (JYT); anaclaudia.marques@unil.ch (ACM)

## Abstract

Pervasive enhancer transcription is at the origin of more than half of all long noncoding RNAs in humans. Transcription of enhancer-associated long noncoding RNAs (elncRNA) contribute to their cognate enhancer activity and gene expression regulation in *cis*. Recently, splicing of elncRNAs was shown to be associated with elevated enhancer activity. However, whether splicing of elncRNA transcripts is a mere consequence of accessibility at highly active enhancers or if elncRNA splicing directly impacts enhancer function, remains unanswered. We analysed genetically driven changes in elncRNA splicing, in humans, to address this outstanding question. We showed that splicing related motifs within multi-exonic elncRNAs evolved under selective constraints during human evolution, suggesting the processing of these transcripts is unlikely to have resulted from transcription across spurious splice sites. Using a genome-wide and unbiased approach, we used nucleotide variants as independent genetic factors to directly assess the causal relationship that underpin elncRNA splicing and their cognate enhancer activity. We found that the splicing of most elncRNAs is associated with changes in chromatin signatures at cognate enhancers and target mRNA expression. We provide evidence that efficient and conserved processing of enhancer-associated elncRNAs contributes to enhancer activity.

## Author summary

Most, if not all, active enhancers are transcribed, giving rise to a plethora of transcripts, including enhancer-associated long noncoding RNAs (elncRNAs). Changes in elncRNA levels impacts cognate enhancer activity. Recently splicing of elncRNA has also been found to associate with enhancer activity. Whether this associations reflects a contribution of elncRNA splicing to increased enhancer activity or else is simply the consequence of increased chromatin accessibility that promotes transcriptional elongation and allows for spurious splicing events remains unknown. We show that natural selection has acted, at the species and population level, to preserve DNA elements required for frequent and efficient elncRNA splicing Importantly, using a genome-wide and unbiased statistical population genomics approach, we demonstrate that elncRNA splicing is associated with cognate enhancer function, contributing to chromatin status and enhancer activity. Our

**Data Availability Statement:** All relevant data are within the manuscript and its Supporting Information files.

**Funding:** ACM was funded by Schweizerischer Nationalfonds zur Förderung der Wissenschaftlichen Forschung (SNF,

PP00P3_179065,). The funders had no role in study design, data collection and analysis, decision to publish, or preparation of the manuscript.

**Competing interests:** The authors have declared that no competing interests exist.

results provides strong evidence that efficient elncRNA splicing contributes to enhancer activity genome-wide.

## Introduction

Classically, enhancers are defined as regulatory DNA elements that positively regulate temporal and spatial expression of their target genes, in *cis*. Transcription activation by enhancers requires transcription factor-dependent recruitment of coactivating complexes and three-dimensional genome rearrangements that bring enhancers into close proximity of their target promoters (reviewed in [1]).

Whereas enhancer activity, as classically defined, is solely determined by its sequence and genomic location, most, if not all, active enhancers are transcribed [2–4]. Initial evidence of this phenomenon emerged from the characterisation of some of the first identified enhancers [5]. Pervasive transcription of active enhancers was later confirmed genome-wide, initially in neurons [3] and macrophages [2] and more recently across a number of human and mouse cell types [6,7]. These studies revealed that enhancer transcription often precedes target promoter activation [2] and is positively correlated with target gene expression [2,3,7]. The majority (>95% [8–10]) of enhancers is transcribed bidirectionally and gives rise to relatively short single exonic RNAs that are non-polyadenylated and unstable, generally referred to as eRNAs [3]. The remaining enhancers transcribe enhancer-associated long noncoding RNAs (elncRNAs [8,9,11]) predominantly in one direction [10]. In contrast with eRNAs, elncRNAs are relatively stable, polyadenylated and can be spliced [10].

Transcription has been found to strengthen cognate enhancer activity through various non-mutually exclusive mechanisms. For example, enhancer transcription facilitates binding of enhancer factors, such as CREBBP [12], or chromatin remodeling complex, including Cohesin and Mediator, that in turn induce RNA-dependent changes in local chromatin conformation [13–15]. Enhancer transcription has also been proposed to regulate the load, pause and release of RNA Polymerase II (RNAPlI) [16,17].

While the contribution of transcription to enhancer function is by now relatively well established, the importance of the transcribed RNAs has only been demonstrated for a handful of anecdotal transcripts (for example, [13,18–21]). In general, the absence of evolutionary constraint at eRNA/elncRNA exons [11] argues that the function of most eRNAs/elncRNAs is unlikely to be encoded within their sequences. Yet a large fraction of elncRNAs is multi-exonic and recently, genome-wide analysis in multiple human cell lines [9] and mouse Embryonic Stem Cells [8] revealed that their splicing positively impacts enhancer function. Enhancers that transcribe multi-exonic elncRNAs are enriched in enhancer-specific chromatin signatures, elevated binding of co-transcriptional regulators, increased local intra-chromosomal DNA contacts, and strengthened *cis*-regulation on target gene expression [8,9]. What remains currently unclear is the causal relationship between elncRNA splicing and enhancer activity. High enhancer activity may promote transcription across splice/donor acceptor site-like sequence, which are common in mammals [22], leading to unregulated/noisy splicing of elncRNAs. Arguing against this is evidence that mutations disrupting elncRNA splicing directly decreased their cognate enhancer activity. For example, removal of the intronic sequence of the locus encoding for the Haunt elncRNA significantly reduced its *cis* regulatory function [23], whereas mutations disrupting the splicing of Blustr, another enhancer-associated lncRNA, were sufficient to negatively impact the expression of its *cis* target [24]. Whether the splicing of most elncRNAs directly impacts enhancer activity, as exemplified by Blustr and

Haunt, or whether it is an inconsequential result of their cognate enhancer's high activity remains unknown.

We used a combination of integrative and population genomics approaches to address this outstanding question. We showed that splicing related motifs within elncRNAs evolved under selective constraint, consistent with their processing being biologically relevant. Using a mediation-based approach, we performed causal inference testing using genetic variants as independent factors to test the underlying causal relationship between elncRNA splicing and their cognate enhancer function. We showed that splicing of most elncRNAs causally contribute to target gene regulation. Furthermore, nucleotide variants that disrupt elncRNA splicing is correlated with significant reduction in cognate enhancer activity, supporting the functional importance of elncRNA splicing.

## Results

### Splicing of elncRNAs is associated with higher enhancer activity

We integrated publicly available RNA sequencing data for human lymphoblastoid cell lines (LCL [25]) with enhancer annotations from the ENCODE consortium [26] to distinguish between predominantly bidirectionally (eRNAs, n = 4433) and unidirectionally transcribed (elncRNAs, n = 564) enhancers active in LCLs (S1 Table and Methods). The transcription profiles of elncRNAs resemble that of promoter-associated lncRNAs (plncRNAs, n = 600) and protein-coding genes (n = 12,070, S1A Fig). In LCLs, most (n = 352, 62.4%) elncRNAs are multi-exonic, consistent with previous analysis in other human cell lines [9].

As described for other human and mouse cells, enhancers that transcribe multi-exonic elncRNAs are more active [8,9]. In particular, compared to enhancers that give rise to either single-exonic elncRNAs or eRNAs, those transcribing multi-exonic elncRNAs have higher levels of chromatin modifications associated with enhancer function (S1B, S1C and S1D Fig), including mono-methylation of Histone 3 Lysine 4 (H3K4me1, S1B Fig), acetylation of Histone 3 lysine 27 (H3K27ac, S1C Fig) and DNase I accessibility (DHSI, S1D Fig). Multi-exonic elncRNA-transcribing enhancers are also enriched for binding of histone acetyl transferase (HAT) P300 (S1E Fig), whose recruitment also distinctly marks active enhancers [27].

Consistent with previous analysis [8,9], relative to all other transcribed enhancers, multi-exonic elncRNA enhancers are enriched in CTFC binding (S1F Fig), significantly enriched at LCL promoter-enhancer loop domains (fold enrichment = 1.3, FDR = $7x10^{-3}$, Methods) and their topologically associating domains (TAD) display a significantly higher frequency of intra-TAD DNA-DNA contacts (p<0.02, two-tailed Mann-Whitney $U$ test, S1G Fig).

In summary, our analysis in LCLs supports that enhancers transcribing multi-exonic elncRNAs are more active, as previously described in other cell types [8,9].

### elncRNA splicing was preserved by purifying selection

The splicing of elncRNAs may be a regulated and conserved process. Alternatively, it can be a mere by-product of inconsequential transcription across splice-site donor/acceptors favoured by increased accessibility at their highly active cognate enhancers. To distinguish between these two possibilities, we first analysed splicing related motifs of elncRNAs. As previously described [8,28,29] the GC contents of exons and introns [30], a determinant of efficient splice-site recognition, of multi-exonic elncRNAs is similar to that of protein-coding genes and promoter-associated plncRNAs, and as observed in mESCs [8] (S2A Fig, two-tailed Mann-Whitney test). Compared to plncRNAs, regions flanking elncRNA splice-sites are enriched in splicing-associated elements, including U1 snRNP binding motifs (S2B Fig, p = 0.05) and a comprehensive (S2C Fig, p = 0.04) [31] and a stringent set (S2D Fig, p = 0.05,

two-tailed Mann-Whitney test)[32] of exonic splicing enhancers (ESEs). This increase in density of splicing-related motifs in elncRNAs support their increased splicing efficiency, as estimated using completed splicing index (coSI), which represents the ratio of reads that span exon-exon splice junctions (excised intron) over those that overlap exon-intron junctions (incomplete excision) [33], relative to plncRNAs (S2E Fig, p = 5x10$^{-11}$, two-tailed Mann-Whitney *U* test), consistent with what was previously reported in mESCs [8].

If splicing of elncRNAs is functionally relevant, one would expect selection to have prevented the accumulation of deleterious mutations in their splicing-associated motifs during evolution. We estimated the nucleotide substitution rate, between human and mouse, for elncRNA splicing-related sequence motifs and compared it to that of neutrally evolving neighbouring ancestral repeats (ARs, Fig 1A). We found that splice donor/acceptor sites, as well as U1 and ESE sites, evolved under significant selective constraint by accumulating fewer substitutions during mammalian evolution than neutrally evolving ARs (p<0.002, permutation test, Fig 1A). Consistent with the observed differences in splicing efficiency, we found that splicing-associated motifs within elncRNAs (including SS, U1 and ESEs, Fig 1B, 1C and 1D), evolved significantly slower (p<2.2x10$^{-16}$, two-tailed Mann-Whitney *U* test) than those within plncRNAs.

We next turned our attention to the evolution of elncRNA splicing during recent human history. Relative to neutrally evolving ARs and plncRNAs, single nucleotide polymorphisms (SNPs) within elncRNA splice sites and splicing motifs were enriched in alleles with low derived allele frequency (DAF < 0.1) (p<0.05, two-tailed Fisher's exact test, Fig 1E), consistent with their evolution under selective constraint. Whereas, the frequency of SNPs with DAF<0.1 that overlap splicing-associated sequences within elncRNAs is statistically indistinguishable from that of protein-coding genes (p = 0.93, two-tailed Fisher's exact test, Fig 1E), we cannot exclude this is in part a consequence of the lack of power to identify differences. Evidence that selection has purged deleterious mutations at elncRNA splicing-associated motifs during mammalian and recent human history, supports the functional relevance of elncRNA processing and argues against it being simply the consequence of increased spurious transcription and splicing at highly active enhancers.

## Disruption of elncRNA splicing decreases target expression

We used genotyping data [34] and gene expression data (Lappalainen et al., 2013) from LCLs derived from 373 healthy individuals to identify nucleotide variants that disrupt elncRNA splicing (Fig 2A). This unbiased approach, which is conceptually analogous to experimentally testing the impact on target expression of introducing point mutations on elncRNA splice sites, allowed us to directly test whether changes in elncRNA splicing impact enhancer activity.

We identified 4 variants (SS variant) that disrupt the splice-donor or acceptor sites of 4 elncRNAs with at least one predicted *cis*-target (S2 Table and Fig 2A and Methods). Since enhancer-promoter interactions rely on local chromosomal looping, enhancers and their associated transcripts often do not target the nearest gene in linear distance [35]. To unbiasedly predict putative elncRNA targets, we identified genes that are jointly associated to the same expression quantitative trait loci (eQTL) variants (Methods), supporting their potential co-regulation [36].

To estimate the impact of SS variant on splicing efficiency, we calculated the Percentage-Spliced-In (PSI) [37] per individual and for each elncRNA splicing event involving the splice donor or acceptor site disrupted by the SS variants (Figs 2B and 2C and S3). PSI measures intron excision by considering spliced reads spanning exon-exon junctions [37]. This is different from the original PSI definition [38,39], which uses the fraction of reads that include or

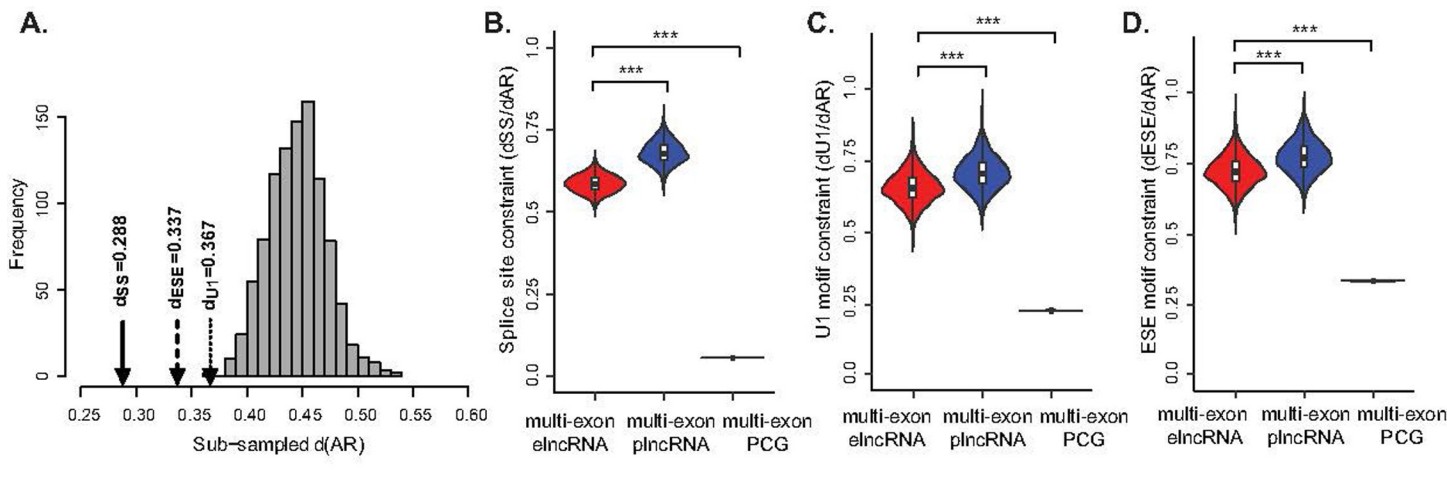

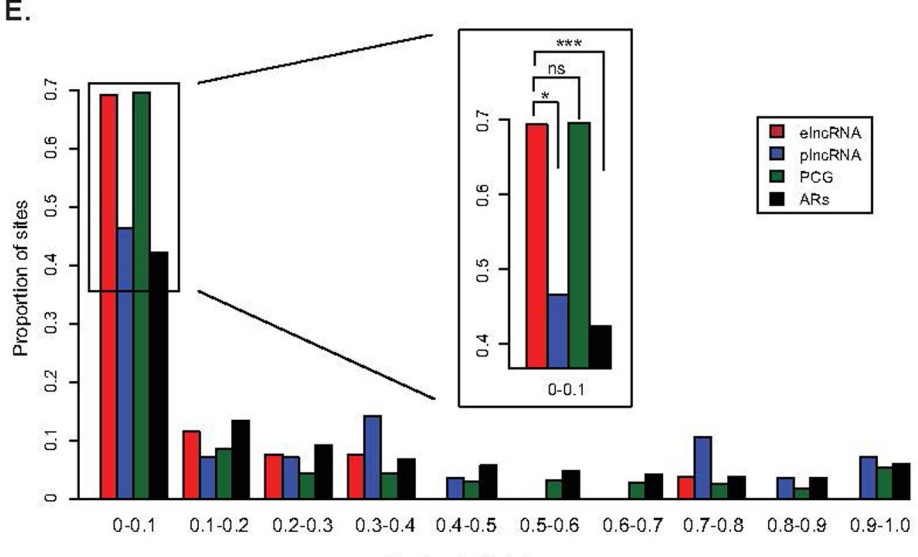

**Fig 1. Splicing of elncRNAs has evolved under purifying selection.** (A) Distribution of pairwise nucleotide substitution rate between human and mouse for 1,000 randomly subsampled sets of local ancestral repeats (ARs) with matching GC-content and size as splicing related motifs (grey). The observed rate observed at splice sites ($d_{SS}$, solid line arrow), ESE ($d_{ESE}$, dash line arrow), and U1 ($d_{U1}$, dotted line arrow) sites within multi-exonic elncRNAs. Distribution of nucleotide substitution rate relative to that of randomly subsampled local ARs ($d_{AR}$) for (B) splice sites, (C) ESEs, and (D) U1s within multi-exonic elncRNA (red), plncRNA (blue) and protein-coding gene (dark green). Differences between groups were tested using a two-tailed Mann-Whitney *U* test. *** $p < 0.001$. (E) Distribution of derived allele frequency (DAF) for single nucleotide variants at splicing motifs (splice sites, ESEs and U1s) within multi-exonic elncRNAs (red), plncRNAs (blue), protein-coding genes (dark green), and ARs (black). The insert illustrates variants with low derived allele frequency (DAF<0.1). Differences between groups were tested using a two-tailed Fisher's exact test. * $p < 0.05$; *** $p < 0.001$; NS $p > 0.05$.

exclude a given exon to measure the efficiency of exon inclusion. We compared the average difference in PSI, as a proxy for change in splicing efficiency, of all affected splicing events between individuals that carry the reference and alternative canonical splice donor/acceptor sites (GT-AG). We found that SS variants are associated with a significant decrease in splicing events involving polymorphic splice donor/acceptor sites (Figs 2B, 2C and 2D and S3). As expected, alongside decreased excision of the affected introns, SS variants are also associated with increased alternative splicing events that involve neighbouring intact splice donor or acceptor sites (Figs 2B and 2C and S4). However, compared to decreased intron excision events observed in individuals carrying the alternative SS variants, the increase in overall alternative splicing

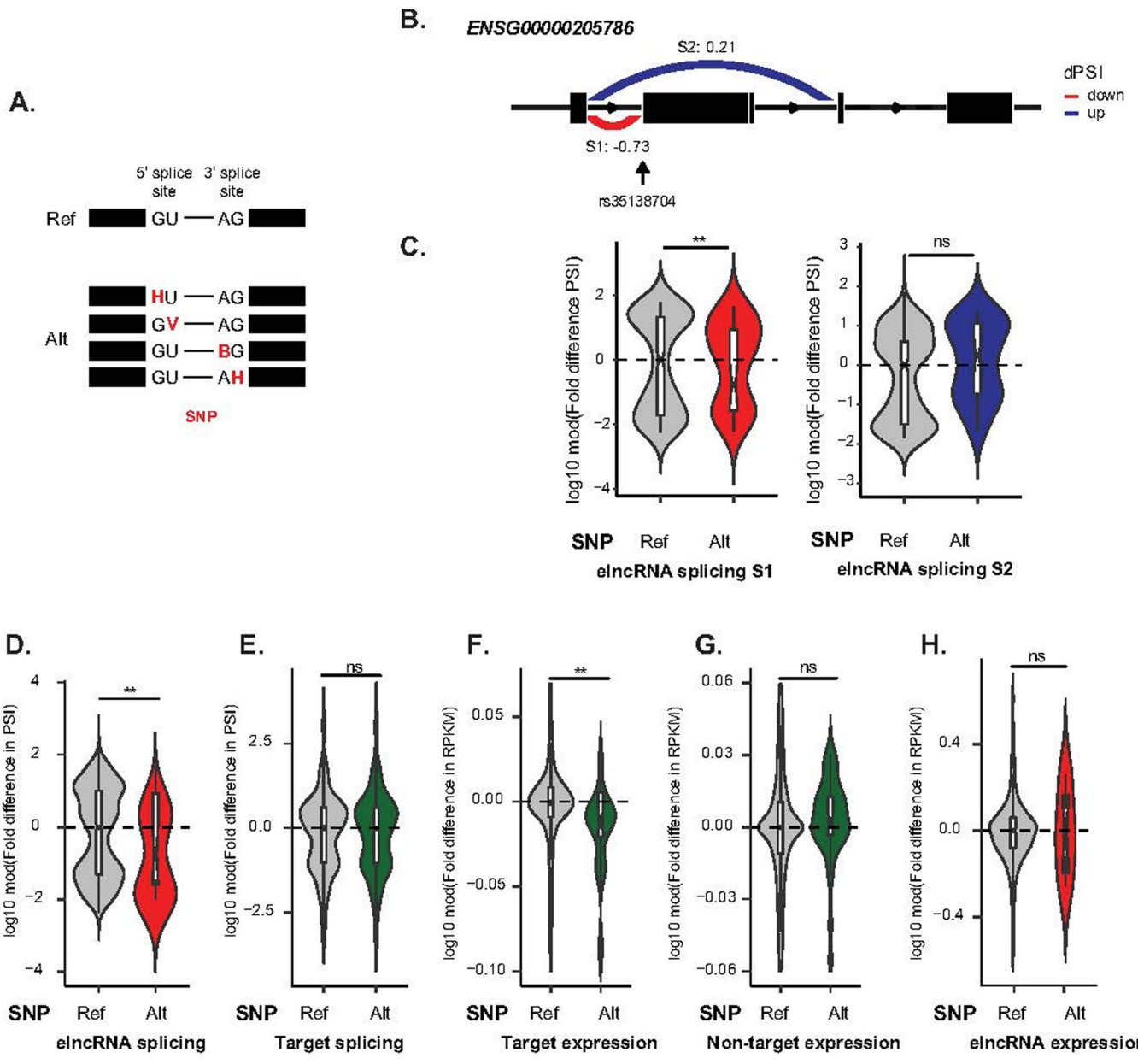

**Fig 2. Disrupted elncRNA splicing impacts *cis*-gene regulation.** (A) Examples of variants that can disrupt elncRNA splicing. In contrast to individuals that carry the reference genome allele at a canonical splice site, those with alternative alleles have at least one variant that disrupt the GU dinucleotide at splice donor site (denoted as HU or GV) or AG dinucleotide at splice acceptor site (denoted as BG or AH). (B) Representation of the differential splicing events between samples with different genotypes for one elncRNA, ENSG00000205786. Median differential splicing (log10 modulus fold difference in Percentage-Spliced-In (dPSI)) of each splicing event is noted next to the arrow. Decreases are represented in red and increases represented in dark blue. (C) Distribution of the log10 modulus fold difference in PSI, relative to the median of samples with reference genotype, between individuals that carry alternative or reference alleles at each corresponding elncRNA splicing event as shown in (B). (D-H) Distribution of the log10 modulus fold difference in splicing (PSI), relative to the median of samples with reference genotype, between individuals that carry alternative or reference alleles at elncRNA splice site of ENSG00000205786, of all directly affected splicing events of (D) the elncRNA and (E) all splicing events of its target protein coding genes (5 targets); as well as fold difference in expression levels (RPKM) of (F) targets, (G) non-targets and (H) the elncRNA. Differences between groups were tested using a two-tailed Mann-Whitney *U* test. * $p < 0.05$; ** $p < 0.01$; *** $p < 0.001$; NS $p > 0.05$.

events, in the same individuals, is 2–8 times smaller and not statistically significant (Figs 2B and 2C and S4). This supports that the presence of SS variants are associated with an overall decrease in splicing. Consistent with a direct role of splicing in the modulation of enhancer function, this

natural mutational study revealed that disruption of elncRNA splicing, which does not impact their protein-coding target splicing (p>0.3, two-tailed Mann-Whitney *U* test, Figs 2E and S4), is associated with significant decrease in target expression (p<0.05, two-tailed Mann-Whitney *U* test, Figs 2F and S4). The association between splice site mutations and gene expression was restricted to predicted elncRNA targets genes, as the expression level of nearby genes was unaffected (p>0.1, two-tailed Mann-Whitney *U* test, Figs 2G and S4). In addition to changes in target protein-coding gene levels, the relative abundance of elncRNAs was also significantly decreased in individuals that carried variants that altered elncRNA splice donor or splice acceptor sites (Figs 2H and S4). This is consistent with the well-established synergy between RNA processing and transcription and the impact of co-transcriptional splicing on gene expression [40,41]. We replicated this mutational study using 89 samples of Yoruba (YRI) population from the Geuvadis dataset and the analysis of these results consistently supports that elncRNA splicing variants impact cognate enhancer function (S5 Fig).

## Causal inference support elncRNA splicing mediates target expression

The results of these mutational studies support the role of splicing in modulating enhancer activity and suggest that this is at least in part a consequence of increased transcription of multi-exonic elncRNAs. However, the interdependence between elncRNA splicing and expression together with the fact that splice site mutations may also affect enhancer factor binding make it difficult to directly use splice site mutations to disentangle the distinct roles of splicing and transcription on enhancer function. Furthermore, although unbiased, splice site mutations were only found for a subset of elncRNAs, restricting the extent of our analysis. To overcome these limitations, we first used multi-variate regression to investigate the relationship between splicing of elncRNAs and the expression of their putative targets genome-wide. First, we found that elncRNA splicing is correlated with target expression levels (S6A Fig) and this correlation is significantly higher than what we found for their proximal non-targets (p<2.2x10^{-16}, two-tailed Mann-Whitney *U* test, S6B Fig). Next, we found by adding elncRNA splicing to its expression significantly, albeit moderately, improved the predicative power of its target gene expression (p = 0.02, two-tailed Mann-Whitney *U* test, S6C Fig). This is in contrast to elncRNA proximal non-targets, whose abundance was not explained by elncRNA splicing (p = 0.8, two-tailed Mann-Whitney *U* test, S6C Fig), supporting that the impact of elncRNA splicing on gene expression is target-specific.

Next, we expanded the number of elncRNA loci considered in our mutational study by identifying SNPs associated with the amount of splicing at multi-exonic elncRNAs (sQTLs) (Fig 3A). To assess the impact of elncRNA splicing on enhancer activity independent of its transcription, we excluded sQTL variants that are also associated with elncRNA expression (eQTLs) from this analysis (Methods) (S7A Fig). The average distance between these variants and elncRNA transcriptional start sites and their associated-enhancer boundaries is 63.7 kb and 72.2 kb, respectively (S7B and S7C Fig), eliminating the potential confounding impact of sQTLs on enhancer factor binding. We identified sQTLs for 25% (88/352) of multi-exonic elncRNAs. Of these, more than half (49, 56%) had at least one sQTL that was also associated with their target expression (80 elncRNA-target pairs, 2,354 joint seQTLs).

Given that quantitative trait variants were identified using a relatively small cohort (373 LCL samples) and that cohort size is an important factor in QTL calling, we reasoned that the relatively small fraction of elncRNAs (~14%, 49/352) with joint seQTL associations is a consequence of our study being under-powered. Using a dataset with nearly 85-times more samples (31,684 blood samples, eQTLGen consortium) [42], we found nearly 80% of elncRNAs (70/88) splicing QTLs are jointly associated to target expression (171 elncRNA-target pairs).

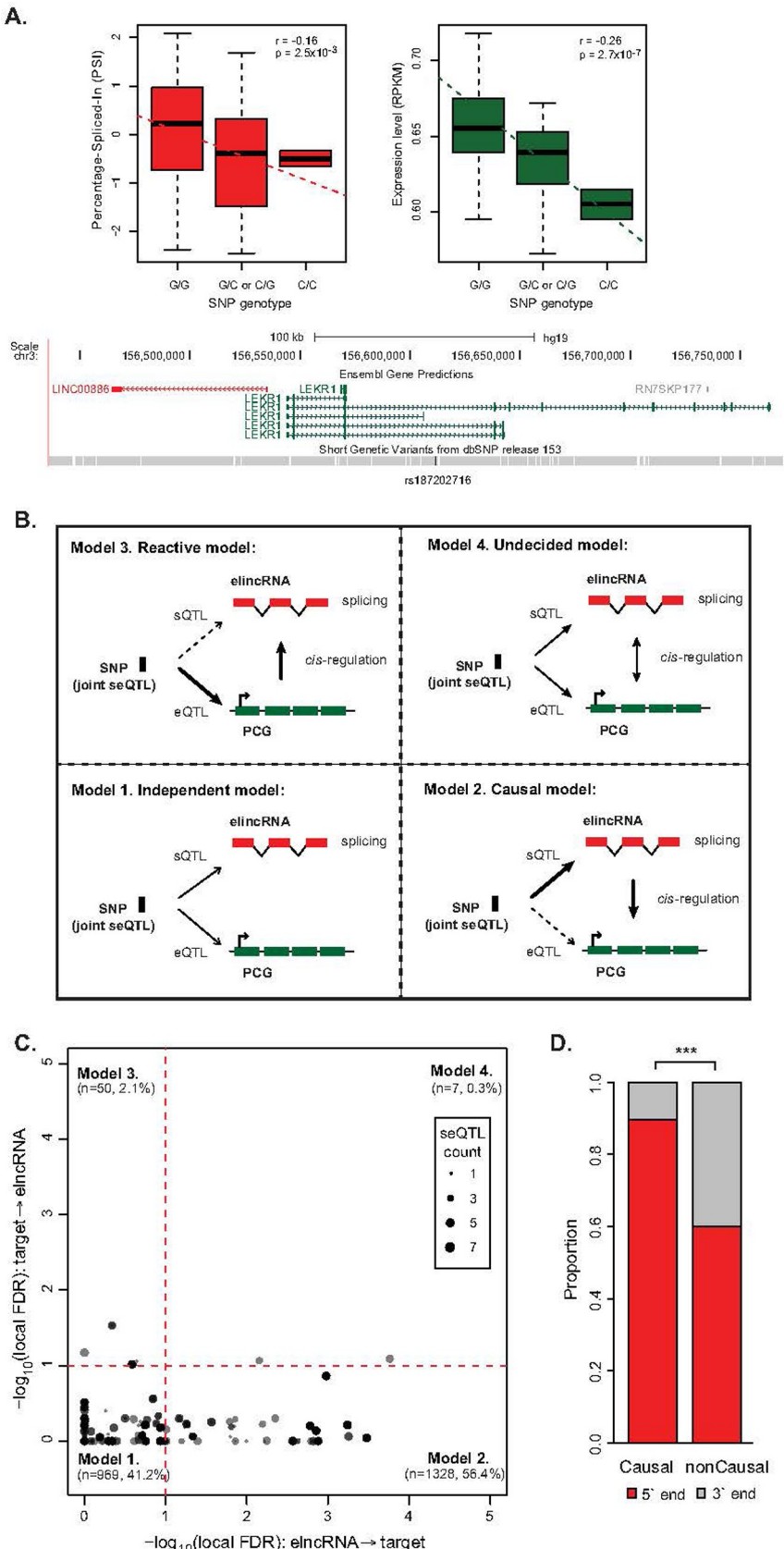

**Fig 3. Impact of elncRNA splicing on *cis*-gene regulation in the human population.** (A) LINC00886 (ENSG00000240875) is a multi-exonic elncRNA whose splicing is associated with genotype of the SNP variant (rs187202716), which is also associated to the expression level of an elncRNA target, LEKR1 (ENSG00000197980). (Top panel) Distribution of the Percentage-Spliced-In (PSI) of elncRNA splicing (LINC00886, red) and target expression (RPKM) (LEKR1, green) in samples across the population that carry different alleles of the SNP variant (rs187202716). Spearman's rho and p-values are shown. (Bottom panel) Genome browser illustrating the genomic positions of the elncRNA (LINC00886, red), its target (LEKR1, green), and their associated SNP (rs187202716, black). (B) Four models of causal inference testing that predict the relationship between joint seQTL variant (black box) associations with the splicing (sQTL) of multi-exonic elncRNAs (red boxes) and the expression level (eQTL) of their target protein-coding genes (green boxes). Schematic representation of the models of joint seQTL associations: (1) the variants are independently associated with elncRNA splicing and target expression (independent model); (2) direct association between the variant and elncRNA splicing mediates the indirect association between that and target expression (causal model); (3) direct association between the variant with target expression mediates the indirect association between that and elncRNA splicing (reactive model); and (4) the causative interaction between elncRNA splicing and target expression is more complex (undecided model). Direct associations are depicted as solid lines and indirect associations as dash lines. (C) Scatterplot depicting causal inference testing local FDR associated with each the four models (as illustrated in B). Number and proportion of joint seQTLs are provided in brackets for each model. Dotted red lines denote significance threshold at local FDR < 0.1. (D) Proportion of joint elncRNA seQTLs causally or non-casually predicted to mediate target gene expression associated with splicing junctions located at the 5´ end (red) or 3´ end (grey) ends of the transcripts. Differences between groups were tested using a two-tailed Fisher's exact test. *** $p < 0.001$.

Compared to the smaller LCL data, target-seQTL associations were considerably higher ($3.45 \times 10^{61}$ times, p<$2.2 \times 10^{-16}$, two-tailed Mann-Whitney *U* test, S7D Fig). In addition, using the larger dataset, target eQTLs that were not identified to be associated with elncRNA splicing by the smaller LCL set (non-seQTLs) were associated to a similar level as those identified to be jointly associated to elncRNA splicing (seQTLs) (p = 0.3, two-tailed Mann-Whitney *U* test, S7E Fig). This corroborates that our analysis is limited by the sample size of the LCL data and we are likely to under-estimate the prevalence of elncRNAs whose splicing contribute to enhancer activity. However, 80% of the expression data used by the eQTLGen consortium was generated using microarray technology which cannot be used to predict splicing frequency. Given this limitation and the restricted access to its sensitive data, we could not carry out detailed analysis using the eQTLGen data and continued to use the LCL dataset.

Different models can account for the joint association between a variant and elncRNA splicing or its target expression (joint seQTL, Fig 3B) [43]. This can occur if the variant is independently associated with elncRNA splicing and target expression (independent model) or as the result of the complex interaction between splicing and expression (undecided model) Alternatively, variant can be associated with the target expression that in turn mediates splicing of elncRNA (reactive model). Finally, cases where the seQTL is associated with elncRNA splicing that in turn mediate target expression (causal model) would support a functional role of elncRNA splicing. For each seQTL, we established the most likely model of association using causal inference analysis [44]. With a false discovery rate of 10%, we found that most associations between seQTLs (Fig 3C) and target expression is causally mediated by elncRNA splicing (n = 1328, 56.4%). These causal seQTLs support the role of splicing for 61% of elncRNAs in the regulation of target expression (1 to 2 targets on average). To assess whether this approach was biased towards the detection of causal seQTLs, we compared the slope and adjusted p-value of the associations between all causal seQTLs and either splicing or expression. As illustrated in S7F Fig, this analysis revealed there is no evidence that stronger sQTLs would favour causal model predictions. Of the remaining variants, most are independently associated with elncRNA splicing and target expression (n = 969, 41.2%), which is likely in part due to the relatively high false negative rates of this type of analysis [44]. To account for the effect of linkage disequilibrium and noise in genotyping data, we repeated the analysis using only seQTLs that are the best (i.e. most significant) variant associated with elncRNA

splicing. Using this conservative set of seQTLs, we identified a similar proportion of elncRNAs (60%) whose splicing is predicted to causally regulate their target expression (S8 Fig), supporting the causal impact of elncRNA splicing on their cognate enhancer function.

We used two independent datasets to assess the robustness of elncRNA target association with sQTL variants we predict to be associated with the splicing of elncRNAs in LCLs. Using a smaller cohort of LCLs (n = 147 [45]), we found a significant association in the same direction between sQTL and target expression for targets of 70% of elncRNAs (45% of variants). A larger fraction of associations (77% of elncRNAs and 52% of variants) could be replicated in a larger cohort of blood samples (n = 31,684 [42]). The difference in size between these two cohorts is likely to explain the difference in replication rate. The association between elncRNA splicing variants and target expression that were replicated have significantly higher effect size relative to non-replicated associations (S9 Fig). Furthermore, LCL-specific effect also likely explains why not all associations can be replicated in the large blood cohort.

Importantly, 90% of seQTL associations that support elncRNA splicing as a mediator of target expression are associated with splicing junctions located at the 5′ end of the transcript, which is consistent with the known synergy between transcription and 5′ end splicing [40,41] (Fig 3D).

### elncRNA splicing impacts local chromatin state

Evidence that variants associated with changes in elncRNA splicing also mediate target gene expression support the direct role of splicing on enhancer function. To assess if elncRNA splicing impacts cognate enhancer function by directly impinging on its activity, we investigated changes in chromatin signatures at elncRNA cognate enhancers. This was possible thanks to the recent release of genome-wide histone modification data, including H3K4me1 and H3K27ac, for a subset of 150 LCL samples [46]. Consistent with a synergistic contribution of splicing to enhancer activity, individuals that carried nucleotide variants that altered elncRNA splice site also had significantly lowered cognate enhancer activity, which we estimated using levels of H3K4me1 and H3K27ac [47] at cognate enhancers (Fig 4A and 4B), supporting that splicing at elncRNA loci impacts local enhancer-associated epigenetic state.

To extend this analysis genome-wide, we next identified nucleotide variants associated with elncRNA splicing and chromatin modification (scQTL, 1,932 and 534 scQTLs involving 30 and 23 elncRNAs, for H3K4me1 and H3K27ac, respectively). We determined the most likely relationship between scQTLs, elncRNA splicing and chromatin signatures using causal inference analysis [44]. We found that a large proportion of scQTL variants (855 (44.3%) and 397 (74.3%) scQTLs for H3K4me1 and H3K27ac, respectively) associated independently to elncRNA splicing and chromatin mark levels. This is expected given the relatively small number of samples available, which has likely reduced the power of this analysis as discussed above. Despite this limitation, we predicted that splicing of a sizable fraction of elncRNAs modulates chromatin state (40% and 48% for H3K4me1 and H3K27ac, respectively, Fig 4C and 4D). These fractions were significantly higher compared to the fraction of elncRNAs whose splicing were predicted to be impacted by chromatin modification changes (3% and 4% for H3K4me1 and H3K27ac, respectively; $p<0.05$, two-tailed Fisher's exact test, Fig 4C and 4D). We provide evidence that splicing a sizable fraction of enhancer-associate transcripts likely to directly impact the activity of their cognate enhancer activity.

## Discussion

We and others have previously shown that splicing of multi-exonic elncRNAs is strongly associated with high enhancer activity in several human and mouse cell lines [8,9]. The direct

## Figure 4.

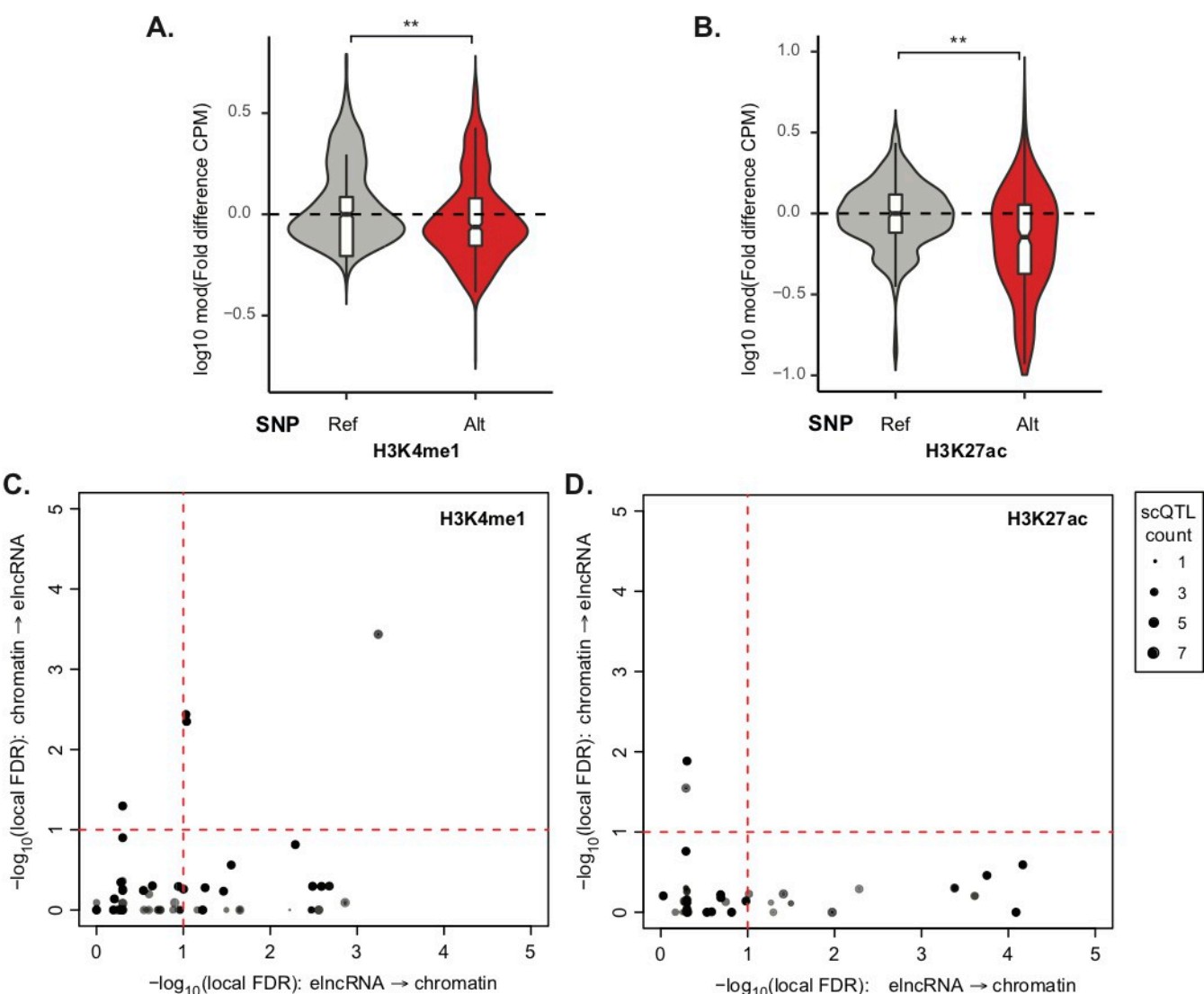

**Fig 4. Impact of elncRNA splicing on cognate enhancer chromatin signatures in the human population.** Distribution of the average log modulus fold difference in chromatin signature, relative to the median of samples with reference genotype, between individuals that carry alternative or reference alleles at elncRNA splice sites, in (A) H3K4me1 and (B) H3K27ac. (C-D) Scatterplot depicting joint elncRNA scQTLs in which elncRNA splicing is causally or non-casually predicted to mediate enhancer chromatin signatures, (C) H3K4me1 and (D) H3K27ac, respectively, using causal inference testing as illustrated using local FDR associated with the four models (as shown in Fig 4B). Dotted black lines denote significance threshold at local FDR < 0.1. Differences between groups were tested using a two-tailed Mann-Whitney $U$ test. ** $p < 0.01$.

contribution of splicing to enhancer activity has been shown for a few candidate elncRNA transcripts [23,24]. However, it remains uninvestigated whether elevated enhancer activity resulting from the splicing of these elncRNAs is a general phenomenon that would explain their previously described genome-wide associations [8,9].

We first aimed to address whether elncRNA splicing is functionally relevant or as a result of transcription through spurious splice sites driven by increased accessibility at highly active enhancers. To gain initial insights into this we first investigated the evolution of splicing-associated motifs, including splice sites, within elncRNAs. We found sequences that support

elncRNA splicing evolved under purifying selection, an evolutionary signature of functionality. Interestingly, splicing related motifs were previously found to be amongst the most highly conserved sequences within lncRNAs [29]. With at least half of all lncRNA transcripts initiating at enhancers [11,48] and a sizable proportion of these undergoing splicing [8,9], it is likely the relative high sequence constraint observed at splicing-associated motifs within lncRNAs [29] can be attributed to enhancer transcribed lncRNAs. Consistent with this, we found splicing-related motifs within the less efficiently processed promoter-associated lncRNAs (plncRNAs) [8] to be less constrained.

We investigated the direct contribution of elncRNA splicing to cognate enhancer activity using a statistical population genomics approach in human lymphoblastoid cell lines (LCLs). We found splicing of at least 60% of elncRNAs causally mediates changes in their target gene expression levels. Interestingly, a sizable proportion (40%) of elncRNA splicing was also predicted to mediate changes in chromatin signatures, including H3K4me1 and H3K27ac, at enhancers. The present analysis is limited by the size of the cohort analysed. Relatively small sample sizes reduce the ability to confidently identify quantitative trait loci [49]. This is illustrated by 1.4-fold increase in the number of seQTL detected when using data from an much larger cohort (eQTLGen [42]) than the one from Geuvadis [25]. We anticipate that the use of larger cohorts should result in an increase in the percentage of elncRNAs for which splicing can be more confidently shown to mediate target expression in *cis*.

Our analysis of elncRNA splice-motif evolution and the impact of elncRNA processing on enhancer function and activity support that elncRNA splicing is unlikely a spurious by-product of enhancer activity. This is consistent with evidence supporting conserved elncRNA splicing and their lack of exonic constraint [8,50], suggesting the contribution of elncRNA splicing to the modulation of enhancer activity go beyond the processing of their mature transcripts.

Whereas the precise molecular mechanisms underlying this association remain currently unknown, a number of mechanisms can support the role of elncRNA splicing on enhancer activity. Splicing has long been known to directly impact the rate of transcription [51]. For example, DNA elements embedded within introns have been shown to contribute to transcriptional regulation [52] and components of the spliceosome can directly enhance RNAPII initiation [53] and transcript elongation [54]. Furthermore, it was recently shown that novel exon splicing events can result in transcription initiation at novel exon proximal promoters, possibly through the recruitment of transcription machinery by splicing factors [55].

The synergy between splicing and transcription may also be influenced by changes in local chromatin environment. For example, splicing pattern changes have been shown to correlate with histone modification and factors that alter chromatin structure, including chromatin signatures at active enhancer regions [56]. On the other hand, tethering of enhancer-associated transcripts at their site of transcription has been shown to alter local chromatin environment [12–14,20]. Chromatin-bound lncRNAs have been recently shown to be enriched in U1 small nuclear ribonucleoprotein (snRNP) RNA-protein complex, a protein essential for the recognition of nascent RNA 5′ splice site and assembly of the spliceosome [57]. The depletion of U1 snRNP reduced the tethering of lncRNAs to chromatin, suggesting a mechanism by which elncRNA splicing can strengthen enhancer activity [57].

The present work provides clues to what might be the molecular mechanism(s) underlying the link between elncRNA splicing and enhancer function. Further experimental work is now needed to dissect the molecular details underlying the role of elncRNA splicing in modulating enhancer activity and investigate the differences between potentially distinct elncRNA classes, for example, by evaluating the consequences on enhancer factor recruitment, chromatin structure and target activity of experimentally perturbing splicing of candidate elncRNA.

## Methods

### Identification of enhancer-associated transcripts

We defined enhancer-associated transcripts by considering all ENCODE intergenic enhancers in human GM12878 lymphoblastoid cell line (LCL. 98,529 enhancers) [58] overlapping DNase I hypersensitive sites [59] and a CAGE cluster [6] in LCLs (n = 4997). We considered all LCL-expressed lncRNAs [60] and Ensembl annotated protein coding genes (version 75). LncRNAs whose 5′ end is within 500bp of a LCL enhancer were considered as being transcribed from the enhancer (n = 564 elncRNAs, S1 Table), and all remaining enhancers were presumed to transcribe eRNAs (n = 4433). Transcripts within 500bp of ENCODE LCL promoters [59] were classified as protein-coding genes (n = 12,070) or promoter-associated lncRNAs (plncRNAs, n = 600) depending on their biotype annotation. We found 62.4% of elncRNAs (n = 352), 60.5% of plncRNAs (n = 363), and 94.4% of protein-coding genes (n = 11,392) to be multi-exonic.

### Read mapping and quantification

For all downloaded data sets, adaptor sequences were first removed from sequencing reads with Trimmomatic (version 0.33) [61] and then aligned to the human reference genome (hg19) using HISAT2 (version 2.0.2) [62]. Since available Geuvadis RNA sequencing and genotype data, LCL CAGE data and enhancer predictions, were all originally mapped to GRCh37 (hg19), our analyses were performed using the same human genome assembly.

### Metagene analysis of elncRNAs

Direction of transcription was assessed with metagene profiles of CAGE reads centered at LCL enhancers and gene transcriptional start sites (TSS) plotted using NGSplot (version 2.4) [63] as in [8]. Enrichment of histone modifications, transcription factor binding, and gene expression levels were assessed using publicly available GM12878 DNase-seq, ChIP-seq and RNA-seq data sets (S3 Table). Metagene profiles of sequencing reads centered at gene TSSs were visualized using HOMER (version 4.7) [64].

### Analysis of chromosomal architecture

Enrichment of enhancer-associated transcripts at LCL loop anchors [65], relative to expectation, was assessed using the Genome Association Tester (GAT, version 1.3.6) [66]. Specifically, loop positional enrichment was compared to a null distribution obtained by randomly sampling 10,000 times (with replacement) segments of the same length and matching GC content as the tested loci within mappable intergenic regions (as predicted by ENCODE [26]). To control for potential confounding variables that correlate with GC content, such as gene density, the genome was divided into segments of 10 kb and assigned to eight isochore bins in the enrichment analysis.

The frequency of chromosomal interactions within topologically associating domains (TADs) was calculated using LCL Hi-C contact matrices [65], as previously described [60].

### Identification of splicing-associated motifs

Density of human exonic splicing enhancers (ESEs), using a set of 238 computationally predicted motifs [31] and a more stringent set of 54 motifs identified across multiple studies [32], were predicted within LCL transcripts as previously described [29]. Specifically, exonic nucleotides (50 nt) flanking splice sites (SS) of internal transcript exons (> 100 nt) were considered in the analysis, after masking the 5 nt immediately adjacent to SS to avoid splice site-associated

nucleotide composition bias [31,67]. Canonical U1 sites (GGUAAG, GGUGAG, GUGAGU) adjacent to 5′ splice sites (3 exonic nt and 6 intronic nt flanking the 5′ SS) were predicted as previously described [68]. FIMO (MEME version 4.12) [69] was used to search for perfect hexamer matches within these sequences.

## GC content

For all LCL expressed genes with at least 2 exons, we computed the GC content for the first exon, remaining exons, and introns for each gene separately, as well as their flanking intergenic sequences of the same length (after excluding the 500 nucleotides immediately adjacent to annotations).

## Splicing efficiency

Transcript splicing efficiency of long non polyA-selected RNA-seq data in GM12878 [59] was estimated by computing the proportion of fully excised introns using bam2ssj (IPSA version 3.3) [70] of transcripts for each gene. We calculated the splicing index, completed splicing index (coSI), which represents the ratio of reads that span exon-exon splice junctions (excised intron) over those that overlap exon-intron junctions (incomplete excision) [33].

## elncRNA conservation across evolution and in humans

To assess selective constraints of transcripts, we estimated their pairwise nucleotide substitution rates. First, pairwise alignment between human and mouse of the different splicing-associated features (including canonical splice sites, exonic splicing enhancers and canonical U1 sites) were separately concatenated for multi-exonic elncRNAs, plncRNAs and protein-coding genes. We used BASEML from the PAML package (version 4.9, REV substitution model [71]) to estimate pairwise nucleotide substitution rates of each splicing motif. To determine whether the splicing motifs have evolved under significant purifying selection, we compared the observed nucleotide substitution rate to that of randomly selected pseudo-splicing motif sequences of the same length and GC content from neighbouring (within 1Mb) neutrally evolving sequences ancestral repeats (ARs, [72]). We repeated this process 1000 times to obtain a distribution of the expected substitution rates.

To assess the conservation of splicing motifs during modern human evolution, we determined their derived allele frequency (DAF), as previously described [50]. We identified common single nucleotide polymorphisms (SNPs) in the human population (dbSNP build 150) mapped within splicing motifs of elncRNAs, plncRNAs and protein-coding genes. DAF was calculated using information on ancestral allele and the frequency of these SNPs in the European population obtained from the 1000 Genomes Project [34]. To determine the expected background distribution, we compared DAF spectrum of the observed splicing motifs to that of randomly selected sequences of the same length and GC content from local ARs.

## Mapping of molecular quantitative trait loci (QTLs)

Expression values (RPKM) of multi-exonic elncRNAs and protein-coding genes in EBV-transformed LCLs derived from 373 individuals of European descent (CEU, GBR, FIN and TSI) were quantified (as described in [60]). The corresponding processed genotypes were downloaded from EBI ArrayExpress (accession E-GEUV-1) [25]. Quantification of alternative splicing events was estimated using LeafCutter (version 1.0) [37]. Single nucleotide polymorphisms (SNPs) located within the same TAD as the genes of interest were tested for association with splicing (sQTLs) and with expression levels (eQTLs) of elncRNAs and protein-coding genes.

Only SNPs with minor allele frequency (MAF) greater than 5% were considered in the QTL analyses. sQTLs and eQTLs were estimated using FastQTL (version 2.184) [73]. To assess the significance of the correlation globally, we permuted the splicing or expression levels of each gene 1000 times and noted the maximum permuted absolute regression coefficient ($r_{max}$). We considered only sQTLs or eQTLs with an observed absolute regression coefficient ($r_{obs}$) greater than 95% of all permuted $r_{max}$ values to be significant [25]. We further performed Benjamini-Hochberg multiple testing correction to estimate FDR ($<5\%$) for all SNPs within the same TAD. Putative protein-coding gene targets of multi-exonic elncRNAs were predicted as those that reside within the same TADs and whose expression levels were associated to the same SNP variant as the expression of the elncRNAs (ie. associated to the same eQTL). We found 88 elncRNAs to have putative protein-coding gene targets and are associated with at least one sQTL (n = 26,741).

Levels of histone modification marks (H3K4me1 and H3K27ac, CPM) that overlap enhancers associated with elncRNAs were downloaded for the 150 European individuals with available chromatin data [46]. To assess the relationship between elncRNA splicing and local histone modifications that mark enhancer elements, we calculated chromatin QTLs (cQTLs) associated with cognate enhancers of elncRNAs in the same way as sQTLs and eQTLs.

## Replication of seQTL associations

Robustness of seQTL associations were assessed using two independent datasets: 1) LCLs (n = 147 [45]) and 2) blood samples (n = 31,684 [42]). After multiple testing correction (Benjamini-Hochberg adjusted p-value $<5\%$) for the number of seQTL variants tested, we considered associations found in the same direction in GTEx LCLs or eQTLgen blood samples as that found in Geuvadis LCLs to be replicated. As only Z-score statistics of eQTL associations is available for download for blood samples [42], which can be transformed to slope of the association using the equation: slope = Z-score / sqrt(2 * alleleFreq * (1-alleleFreq) * (N + Z-score^2)) (N = sample size, alleleFreq = allele frequency of variant). However, since we do not have information on the allele frequency of the genetic variants, we directly used Z-score for these samples in our analysis.

## Impact of genetic variation at elncRNA splice sites on cis-gene expression

We considered all SNPs located at elncRNA splice sites (SS variants) and measured the fold difference for all elncRNA splicing events (using Percentage-Spliced-In (PSI) by LeafCutter [37]) affected by the SS variants between individuals that carry the reference or alternative alleles of these variants (S2 Table). Similarly, fold difference in elncRNA steady state abundance, as well as that of their putative target, and chromatin marks at their cognate enhancer elements were also assessed. In each case, we show the distribution of the fold difference in PSI or RPKM or CPM for each individual relative to the median PSI or RPKM or CPM in individuals with the reference genotype (Figs 2C–2H, 4A and 4B and S3, S4 and S5). Intron-centric splicing efficiency as measured by LeafCutter is visualized by LeafViz [37].

## Causality inference between elncRNA splicing and nearby protein-coding gene expression

To infer the causal relationship between elncRNA splicing and putative target gene expression, we focused on QTLs that are associated with both splicing of elncRNAs (sQTL) and their putative target gene abundance (eQTL), and we refer to these variants as joint seQTLs. elncRNA sQTL variants that were also associated with elncRNA expression level or splicing of their

putative *cis*-target genes were excluded from the analysis (n = 21,650 out of 26,741) (S7A Fig). In total, we found 49 elncRNAs with seQTLs shared with 80 target genes.

For all triplets of seQTL–elncRNA splicing–target gene expression (n = 2,349), we performed causal inference testing using a Bayesian approach as implemented by Findr [44] by testing the models: (1) the independent model where seQTL variants are independently associated with elncRNA splicing and target gene expression; (2) the causal model with elncRNA splicing as the molecular mediator of gene expression; (3) the reactive model where gene expression mediates elncRNA splicing; and (4) the undecided model where causative interaction between elncRNA splicing and target gene expression is more complex [43].

The same causal inference testing was performing between elncRNA splicing (sQTL) and chromatin marks (cQTL) at their cognate enhancers by identifying scQTLs. We found 30 and 23 elncRNAs whose splicing-QTL is also associated with H3K4me1 (n = 1391 triplets) and H3K27ac (n = 532 triplets) marks, respectively, at their cognate enhancers.

### Linear regression

We used multi-variable regression to assess whether elncRNA splicing contributes to their cognate enhancer activity by comparing the difference in the proportion of variance in target expression is explained by adding elncRNA splicing to the model (Target-expression ~ elncRNA-expression compared to Target-expression ~ elncRNA-expression + elncRNA-splicing).

### Statistical tests

All statistical analyses were performed using the R software environment for statistical computing and graphics [74]. When multiple comparisons are made, we report the highest p-value of all comparisons.

### Supporting information

**S1 Fig. Multi-exonic elncRNAs are transcribed from highly active enhancers.** (A) Metagene plots of CAGE reads centered at transcription initiation regions (TIRs) of eRNAs, and promoters (estimated as -500bp to annotated gene TSS) of elncRNAs, plncRNAs and protein-coding genes (PCGs). Sense (red) and antisense (blue) reads denote those that map to the same or opposite strand, respectively, as the direction of their cognate TIRs. Metagene plots and distribution (figure insets) of (B) H3K4me1, (C) H3K27ac, (D) DNase I hypersensitive sites (DHSI), (E) P300 and (F) CTCF ChIP-seq reads in LCLs at promoters of multi-exonic (red) and single-exonic (grey) elncRNAs, and eRNAs (yellow). (G) Distribution of the average amount of DNA:DNA contacts within LCL TADs containing multi-exonic (red) and single-exonic (grey) elncRNAs and eRNAs (yellow). Differences between groups were tested using a two-tailed Mann-Whitney *U* test. * $p < 0.05$; *** $p < 0.001$.
(TIFF)

**S2 Fig. Splicing of elncRNAs has evolved under purifying selection.** (A) Distribution of GC-content across exons and introns of elncRNAs (red), plncRNAs (blue), and protein-coding genes (green). Distribution of the density of predicted (B) U1 spliceosome RNAs (snRNPs) and exonic splicing enhancers (ESEs), using (C) a comprehensive (n = 238) and (D) a smaller but more stringent set (n = 54) of predicted ESE motifs, within multi-exonic elncRNAs (red), plncRNAs (blue) and protein-coding genes (green). (E) Distribution of the completed splicing index (coSI) for multi-exonic elncRNAs (red), plncRNAs (blue) and protein-coding genes (green). Differences between groups were tested using a two-tailed Mann-Whitney *U* test. *

$p < 0.05$; ** $p < 0.01$; *** $p < 0.001$; NS $p > 0.05$.
(TIFF)

**S3 Fig. Disrupted elncRNA splicing impacts *cis*-gene regulation.** Distribution of the log10 modulus fold difference in splicing (PSI), relative to the median of samples with reference genotype, between individuals that carry alternative or reference alleles at elncRNA splice site of (A) ENSG00000250903, (B) ENSG00000227388 and (C) XLOC_013468 of all directly affected splicing events of the elncRNA and all splicing events of their target protein coding genes (1, 7, and 1 targets, respectively); as well as fold difference in expression levels (RPKM) of targets, non-targets and the elncRNA. Differences between groups were tested using a two-tailed Mann-Whitney *U* test. * $p < 0.05$; ** $p < 0.01$; *** $p < 0.001$; NS $p > 0.05$.
(TIFF)

**S4 Fig. Disrupted elncRNA splicing impacts *cis*-gene regulation.** (Top panels) Representation of the differential splicing events between samples with different genotypes for elncRNAs, (A) ENSG00000250903, (B) ENSG00000227388 and (C) XLOC_013468. Median differential splicing (log10 modulus fold difference in Percentage-Spliced-In (dPSI)) of each splicing event is noted next to the arrow. Decreases are represented in red and increases represented in dark blue. (Bottom panels) Distribution of the log10 modulus fold difference in PSI, relative to the median of samples with reference genotype, between individuals that carry alternative or reference alleles at each corresponding elncRNA splicing event as shown in the top panels. Differences between groups were tested using a two-tailed Mann-Whitney *U* test. * $p < 0.05$; ** $p < 0.01$; *** $p < 0.001$; NS $p > 0.05$.
(TIFF)

**S5 Fig. Disrupted elncRNA splicing impacts *cis*-gene regulation across YRI population.** Distribution of the median log10 modulus fold difference, relative to the median of samples with reference genotype, between individuals of the Yorubin (YRI) population that carry alternative or reference alleles at elncRNA splice sites, in splicing (PSI) of multi-exonic elncRNAs and target protein coding genes; target and non-target gene expression levels (RPKM); and elncRNA expression. Differences between groups were tested using a two-tailed Mann-Whitney *U* test. * $p < 0.05$; ** $p < 0.01$; *** $p < 0.001$; NS $p > 0.05$.
(TIFF)

**S6 Fig. elincRNA splicing contributes to target expression levels.** (A) elncRNA splicing is significantly correlated (Spearman's test) with their *cis*-target expression levels, as illustrated using ENSG00000227036 (elncRNA) and ENSG00000125398 (target protein-coding gene). (B) Distribution of Spearman's correlation between elncRNA splicing and either their targets or non-targets. (C) Distribution of adjusted R square comparing between two regression models that use elncRNA expression and its splicing to predict expression of their targets or non-targets: Models 1: PCG-expression ~ elncRNA-expression; Model 2: PCG-expression ~ elncRNA-expression + elncRNA-splicing. Differences between groups were tested using a two-tailed Mann-Whitney *U* test. * $p < 0.05$; ** $p < 0.01$; *** $p < 0.001$; NS $p > 0.05$.
(EPS)

**S7 Fig. Impact of elncRNA splicing on *cis*-gene regulation in the human population.** (A) Distribution of the absolute slope (left panel) and the -log10 of the adjusted p-value (right panel) of elncRNA sQTLs that are excluded (grey) or included (green) in analysis due to their significant associations with elncRNA expression, respectively. Distribution of distance between genomic coordinate of seQTL and position of their associated elncRNA annotated transcriptional start sites (B) and enhancers (C). (D) Distribution of elncRNA sQTLs that are

also associated with a *cis*-target expression (seQTL) as identified using a LCL population with 373 samples (Geuvadis) or blood samples (eQTLGen) with 31,684 samples. (E) Distribution of elncRNA seQTLs and non-seQTLs (elncRNA sQTLs not associated with target expression in LCL population) associations as identified in LCL Geuvadis population or in blood eQTLGen samples. (F) Distribution of the absolute slope (left panel) and the -log10 of the adjusted p-value (right panel) of elncRNA sQTL association (red) and target eQTL association (green) for all causal seQTLs. Differences between groups were tested using a two-tailed Mann-Whitney *U* test. $^*$ $p < 0.05$; $^{**}$ $p < 0.01$; $^{***}$ $p < 0.001$; NS $p > 0.05$.
(TIFF)

**S8 Fig. Impact of elncRNA splicing on chromatin signature associated with enhancer activity in the human population.** Scatterplot depicting the best joint elncRNA seQTLs (the most significant variant associated with elncRNA splicing) used to infer relationship between elncRNA splicing and *cis*-target expression levels using causal inference testing, as illustrated using local FDR associated with the four models (as illustrated in Fig 4B). Dotted black lines denote significance threshold at local FDR < 0.1.
(TIFF)

**S9 Fig.** Distribution of the absolute slope or z-score (left panels) and adjusted p-value (right panels) of elncRNA target eQTL associations that can be replicated (after multiple-testing correction <5% and in the same association direction, green) or not (grey) using LCL samples from GTEx (A) and blood samples from eQTLgen (B).
(TIFF)

**S1 Table. Genomic position of single- and multi-exonic LCL elncRNAs (hg19) and their respective number of exons.**
(XLSX)

**S2 Table. Single nucleotide polymorphisms (SNPs) at multi-exonic elncRNA splice sites and their respective targets, as well as SNPs jointly associated to both elncRNA and target expression levels.** SNPs are denoted in the format snp_chr_position.
(XLSX)

**S3 Table. Publicly available datasets used in the analysis.**
(XLSX)

## Acknowledgments

We thank members of the Marques group for valuable comments and discussion. We thank Zoltán Kutalik and Diogo Ribeiro for discussion on population genomics analysis; Olivier Delanau for early access to chromatin mark quantification.

## Author Contributions

**Conceptualization:** Jennifer Yihong Tan, Ana Claudia Marques.

**Data curation:** Jennifer Yihong Tan.

**Formal analysis:** Jennifer Yihong Tan.

**Funding acquisition:** Ana Claudia Marques.

**Investigation:** Jennifer Yihong Tan.

**Methodology:** Jennifer Yihong Tan, Ana Claudia Marques.

**Project administration:** Ana Claudia Marques.

**Visualization:** Jennifer Yihong Tan.

**Writing – original draft:** Jennifer Yihong Tan, Ana Claudia Marques.

**Writing – review & editing:** Ana Claudia Marques.

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
