## [Decision Letter · Decision Letter 0]

16 Nov 2020

Dear Dr. Marques,

Thank you very much for submitting your manuscript "The activity of human enhancers is modulated by the splicing of their associated lncRNAs." for consideration at PLOS Computational Biology.

As with all papers reviewed by the journal, your manuscript was reviewed by members of the editorial board and by several independent reviewers. In light of the reviews (below this email), we would like to invite the resubmission of a significantly-revised version that takes into account the reviewers' comments.

We cannot make any decision about publication until we have seen the revised manuscript and your response to the reviewers' comments. Your revised manuscript is also likely to be sent to reviewers for further evaluation.

It is absolutely mandatory that the concerns of the Reviewer 2 will be resolved to his/her satisfaction.

Sincerely,

Ilya Ioshikhes

Associate Editor

PLOS Computational Biology

William Noble

Deputy Editor

PLOS Computational Biology

Reviewer's Responses to Questions

**Comments to the Authors:**

Reviewer #1: I praise and thank the authors for addressing my concerns in a mostly satisfactory way. My remaining requests for revision aim to improve the clarity of the description of the splicing analyses and results, namely by the correction of some misleading statements. In the way the manuscript is currently written, there is an apparent mix-up of concepts related to splicing efficiency and alternative splicing quantification.

Minor essential revisions:

1. The splicing community commonly uses PSI (percent spliced-in) as a metric for quantification of alternative splicing that is computed based on the profiling of alternative exon-exon (i.e. spliced) junctions. This is, to my knowledge, how LeafCutter operates, using split reads (i.e. reads covering exon-exon junctions) to uncover alternative intron-excision options and the PSI values provided therein and used in Figures 3, S3 and S4 are ratios involving only spliced transcripts. The authors chose to also call PSI to a splicing index (originally named coSI in the Tilgner 2012 paper) that involves a ratio between (split) reads spanning exon-exon junctions and (unsplit) reads covering exon-intron junctions, as a metric for splicing efficiency. Although those two quantifications are not totally independent (the more efficient the splicing, the more exons get included), their conceptual difference is very important for our understanding of splicing efficiency and mechanisms of alternative splicing regulation, which are also different concepts. The two PSI metrics need to be clearly disambiguated.

2. Exon “skipping” means exon “exclusion”, not “inclusion”. This would require correction of the newly written lines 25-28 of page 10, as they currently convey contradictory messages, and of whatever other sentence in the manuscript where “skipping” may be misused. However, even after such correction, those two sentences are misleading. The fact that some log-fold differences in PSI for the alternative allele are positive does not mean that the variants can promote more inclusion, as their PSIs are compared to a median reference genotype sample and not the reference genotype in the same individuals. In other words, those individuals with positive log-fold differences could have higher-than-average PSIs if they had a reference genotype and therefore still have a decrease from a much-higher-than-median to a slightly-higher-than-median PSI with the change in genotype. As the comparisons are not “paired” and therefore do not control for the trans-regulation of splicing, statements can only be made about the distributions as a whole (i.e. their average behaviours) and not about individual values.

Optional/discretionary edits:

3. I disagree with the authors’ statements, not justified in the rebuttal, that violin plots have a “less intuitive” interpretation and boxplots are “simpler and easier to interpret”. It is hard to understand how a plot that conveys a smoothed histogram of the full distribution can be less intuitive, particularly when it can be plotted behind a boxplot to convey quartiles. The choice of less informative plots is even more intriguing in a Computational Biology journal. Having said this, it is an issue that does not affect this manuscript’s suitability for publication. My remark only intends to promote a good practice that would bring extra scrutinisability to the results of this work.

4. Plots involving p-values (i.e. significance) in Supplementary Figure S7 would be more easily interpretable by the readers if -log10 (instead of just log10) was used. Higher significance should show up higher in the plots. I apologize for having overlooked this in the first review.

Reviewer #2: The authors have revised their manuscript, but the results lack novelty and the data do not compellingly support the conclusions drawn.

Regarding novelty, Figure 1 of the manuscript is virtually identical in content to Figure 2 of a paper recently published by the authors (https://www.life-science-alliance.org/content/3/4/e202000663), showing essentially the same result in a slightly different system. So this result is purely confirmatory and should perhaps be a supplementary figure. Figure 2 shows an evolutionary result that supports that splicing of elncRNAs is under purifying selection, but provides no evidence of what the function of this splicing might be, so is novel but plays a minor role in advancing the authors' overall argument.

Thues, the main direct evidence to support the authors' main conclusion that splicing of elncRNAs increases enhancer activity is still in Figure 3. And still the data are extremely weak. The analysis is still based on an n of 4, and the effects are extremely small. Alternate possibilities such as that the elncRNA expression rather than splicing modulates enhancer activity are still not excluded. Overall, I don't feel that the authors have improved the manuscript in a substantive way. The data supporting the main conclusions are still not compelling, and the novelty is compromised by their recent publication, cited above.

**Have all data underlying the figures and results presented in the manuscript been provided?**

Reviewer #1: Yes

Reviewer #2: Yes

PLOS authors have the option to publish the peer review history of their article (what does this mean?). If published, this will include your full peer review and any attached files.

Reviewer #1: No

Reviewer #2: No
---

## [Decision Letter · Decision Letter 1]

27 Apr 2021

Dear Dr. Marques,

Thank you very much for submitting your manuscript "The activity of human enhancers is modulated by the splicing of their associated lncRNAs." for consideration at PLOS Computational Biology.

As with all papers reviewed by the journal, your manuscript was reviewed by members of the editorial board and by several independent reviewers. In light of the reviews (below this email), we would like to invite the resubmission of a significantly-revised version that takes into account the reviewers' comments. In particular, we suggest that you consider the comment #1 of Reviewer #4, and test the hypothesis in an experimental setting.

We cannot make any decision about publication until we have seen the revised manuscript and your response to the reviewers' comments. Your revised manuscript is also likely to be sent to reviewers for further evaluation.

Sincerely,

Yi Xing

Guest Editor

PLOS Computational Biology

William Noble

Deputy Editor

PLOS Computational Biology

Reviewer's Responses to Questions

**Comments to the Authors:**

Reviewer #1: I praise and thank the authors for their efforts in addressing my minor concerns.

Unfortunately, they appear to keep struggling with the concepts and terminology associated with alternative splicing and its quantification. The Percentage-Spliced-In (PSI), as the name indicates, measures exon INCLUSION, and not “exclusion” (that would be 1-PSI or 100%-PSI) as stated in the paper. It is very important that the definitions are accurate and in line with the usual interpretation, so that the readers are not (unintentionally) misled.

Similarly, the confusion between PSI and coSI appears to remain. In Figure 2B,C and Figure S3, the authors assign differences in PSI to S1 and S2, representing single exon-exon junctions. Based on the PSI definition, it is unclear which alternative exons S1 and S2 refer to or why distributions in actual PSI (instead of fold differences) are not compared. The meaning of ratios between exonic and intronic sequences (indeed putatively associated with splicing efficiency) appear to be confused with ratios between alternative exonic sequences (standard alternative splicing quantifications).

Along the same lines, what “other alternative splicing events” (page 10, lines 20-21), other than exon skipping, can SS variants promote that a reader can see in Figures 2B,C and S3? What does an “overall decrease in splicing events” mean? Does it mean that that more exons are skipped or more introns are retained?

I strongly recommend that the authors do not underestimate the importance of this issue and thoroughly revise these definitions and quantifications with help from an alternative splicing expert. Until this is done properly, the results from this work, even if sound, will remain hardly intelligible for most readers and, in my view, not fully ready for publication.

Reviewer #4: I appreciate the authors' efforts to address the issues of novelty and evidence for their conclusions, as well as other points raised. This new version of the manuscript is significantly stronger. However, there are a few specific points that should be developed further before publication.

1. Direct evidence of causality. The main evidence supporting the authors’ conclusion that splicing of elncRNAs increases enhancer activity is still aggregated in a very small number of loci (n=4). I agree with the authors that this in silico experiment is “analogous” to experimentally disrupting splice sites of selected elncRNA candidates. However, I do not believe both approaches are equivalent. At this point, I suggest the authors actually test their hypothesis in an experimental setting. They could pick a few candidates to perturbate splice sites elncRNA with punctual mutations that create weaker or stronger splice sites and evaluate the consequences on enhancer’s activity and chromatin structure.

2. Link between splicing and increased transcription. The authors mentioned that they only considered variants associated with elncRNA splicing but not with its expression in Figs. 3 and 4, but do not show the data (sorry in case I missed it). I think that showing that the total elncRNA expression does not change is as crucial as showing that the splicing of the elncRNA does change.

**Have all data underlying the figures and results presented in the manuscript been provided?**

Reviewer #1: Yes

PLOS authors have the option to publish the peer review history of their article (what does this mean?). If published, this will include your full peer review and any attached files.

Reviewer #1: No

Reviewer #4: No

**Have the authors made all data and (if applicable) computational code underlying the findings in their manuscript fully available?**

Reviewer #4: Yes
---

## [Decision Letter · Decision Letter 2]

27 Oct 2021

Dear Dr. Marques,

Thank you very much for submitting your manuscript "The activity of human enhancers is modulated by the splicing of their associated lncRNAs." for consideration at PLOS Computational Biology.

As with all papers reviewed by the journal, your manuscript was reviewed by members of the editorial board and by several independent reviewers. In light of the reviews (below this email), we would like to invite the resubmission of a significantly-revised version that takes into account the reviewers' comments.

In particular, we would require you to follow the advice on toning down your conclusions as suggested by Reviewer 4 and treat this work as more hypothesis generating than conclusive.

We cannot make any decision about publication until we have seen the revised manuscript and your response to the reviewers' comments. Your revised manuscript is also likely to be sent to reviewers for further evaluation.

Sincerely,

Ilya Ioshikhes

Deputy Editor

PLOS Computational Biology

William Noble

Deputy Editor

PLOS Computational Biology

Reviewer's Responses to Questions

**Comments to the Authors:**

Reviewer #1: I praise and thank the authors for carefully addressing my concerns.

Reviewer #4: The authors have done a great job addressing my comment on the link between splicing and increased transcription. I appreciate the new Supplementary figure which is quite strong. In addition, I of course understand that disrupting splice sites of selected elncRNA candidates would be a hard experiment to set up and might not be possible in LCL cells. However, I still believe that the evidence in this manuscript to support the main conclusion of causality (which is the take home message and differentiates this paper from previous literature) is not conclusive. I believe that experimental evidence is necessary to support the conclusion that splicing of elncRNAs increases enhancer activity and several labs around the world would be happy to collaborate in such efforts. If the paper should be published without experimental data, I suggest to tone down the conclusions and acknowledge this issue.

**Have the authors made all data and (if applicable) computational code underlying the findings in their manuscript fully available?**

Reviewer #1: Yes

Reviewer #4: None

PLOS authors have the option to publish the peer review history of their article (what does this mean?). If published, this will include your full peer review and any attached files.

Reviewer #1: No

Reviewer #4: No
---

## [Decision Letter · Decision Letter 3]

5 Dec 2021

Dear Dr. Marques,

We are pleased to inform you that your manuscript 'The activity of human enhancers is modulated by the splicing of their associated lncRNAs.' has been provisionally accepted for publication in PLOS Computational Biology.

Please correct the text as requested by reviewer on the same occasion.

Best regards,

Ilya Ioshikhes

Deputy Editor

PLOS Computational Biology

William Noble

Deputy Editor

PLOS Computational Biology

Reviewer's Responses to Questions

**Comments to the Authors:**

Reviewer #4: The authors have addressed my concerns and I support the publication of this manuscript. I would suggest the authors to test their hypothesis in an experimental setting when possible. Also, in page 3, I would suggest to use "evidence" instead of "strong evidence".

**Have the authors made all data and (if applicable) computational code underlying the findings in their manuscript fully available?**

Reviewer #4: Yes

PLOS authors have the option to publish the peer review history of their article (what does this mean?). If published, this will include your full peer review and any attached files.

Reviewer #4: No

---

## [Editor Report · Acceptance letter]

6 Jan 2022

PCOMPBIOL-D-20-01247R3 

The activity of human enhancers is modulated by the splicing of their associated lncRNAs.

Dear Dr Marques,

I am pleased to inform you that your manuscript has been formally accepted for publication in PLOS Computational Biology. Your manuscript is now with our production department and you will be notified of the publication date in due course.

With kind regards,

Katalin Szabo
